

# A psychophysical evaluation of techniques for Mooney image generation

Lars C. Reining[1] and Thomas S. A. Wallis[1,2]

[1] Technical University of Darmstadt, Darmstadt, Germany
[2] Center for Mind, Brain and Behavior (CMBB) Universities of Marburg, Giessen and Darmstadt, Marburg, Giessen and Darmstadt, Germany

## ABSTRACT

Mooney images can contribute to our understanding of the processes involved in visual perception, because they allow a dissociation between image content and image understanding. Mooney images are generated by first smoothing and subsequently thresholding an image. In most previous studies this was performed manually, using subjective criteria for generation. This manual process could eventually be avoided by using automatic generation techniques. The field of computer image processing offers numerous techniques for image thresholding, but these are only rarely used to create Mooney images. Furthermore, there is little research on the perceptual effects of smoothing and thresholding. Therefore, in this study we investigated how the choice of different thresholding techniques and amount of smoothing affects the interpretability of Mooney images for human participants. We generated Mooney images using four different thresholding techniques, selected to represent various global thresholding methods, and, in a second experiment, parametrically varied the level of smoothing. Participants identified the concepts shown in Mooney images and rated their interpretability. Although the techniques generate physically-different Mooney images, identification performance and subjective ratings were similar across the different techniques. This indicates that finding the perfect threshold in the process of generating Mooney images is not critical for Mooney image interpretability, at least for globally-applied thresholds. The degree of smoothing applied before thresholding, on the other hand, requires more tuning depending on the noise of the original image and the desired interpretability of the resulting Mooney image. Future work in automatic Mooney image generation should pursue local thresholding techniques, where different thresholds are applied to image regions depending on the local image content.

## INTRODUCTION

In the realm of classical vision science, the majority of stimuli are typically characterized by their simplicity and artificiality. Frequently, a preliminary study is undertaken to identify the most suitable parameters, which are then utilized in the generation of new stimuli. With the continual expansion of the internet's image database, more modern approaches on the other hand, use natural images or manipulated natural images as stimuli. However,

Corresponding authors
Lars C. Reining,
lars.reining@stud.tu-darmstadt.de
Thomas S. A. Wallis,
thomas.wallis@tu-darmstadt.de

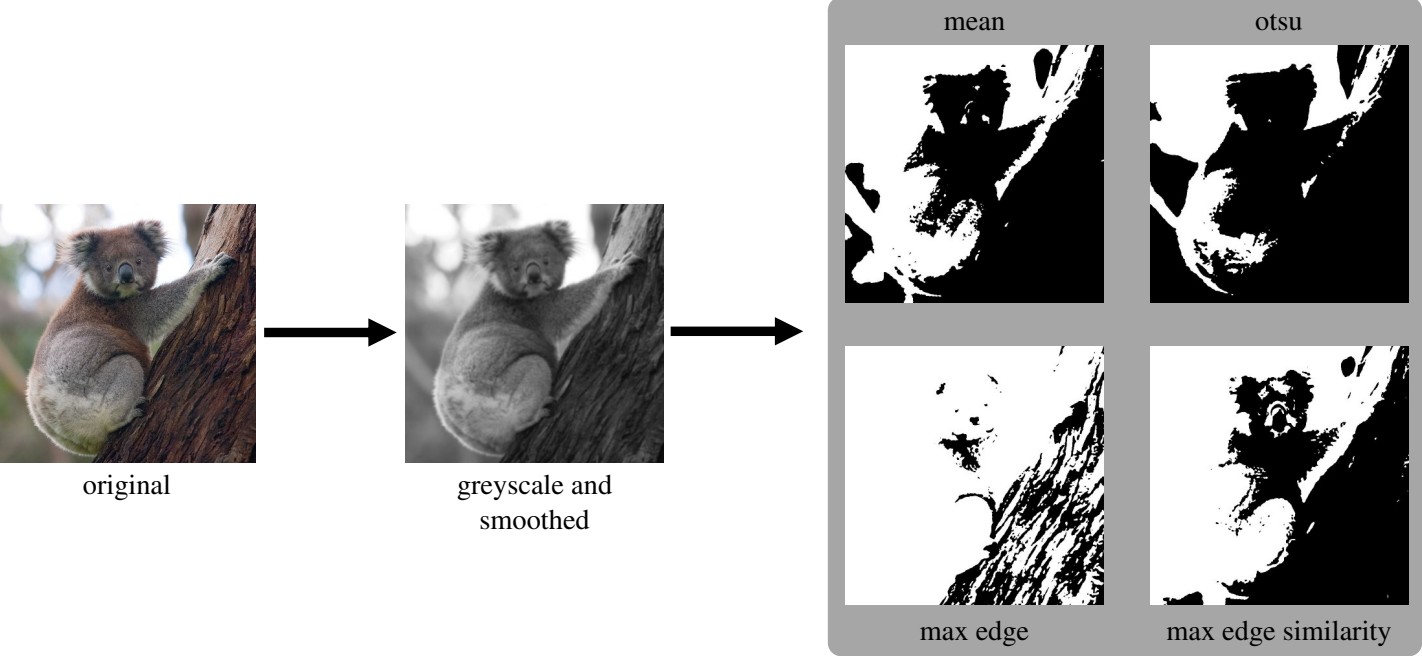

**Figure 1 Mooney image generation pipeline applied to one example image from the THINGS dataset.** In the first step the images are converted to grayscale and smoothed with a gaussian filter. In the second step four different image thresholding techniques are applied to the smoothed image. The template image is taken from the THINGS dataset (*Hebart et al., 2019*), which is shared under a CC-BY-4.0 license. The following pictures were modified according to the description found in the methods section.

due to the inherent diversity of underlying images, merely applying repetitive manipulations may not yield stimuli which are useful in testing hypothesizes and comparing theories. As a result, various types of image-based stimuli continue to be manually crafted to ensure their interesting nature.

One classical example of a stimulus that has been used in vision science and is typically manually crafted are "Mooney images" (named for *Mooney, 1957*). Mooney images are a special kind of two-tone images, *i.e.*, images where each pixel is either completely black or white depending on its original luminance (for examples see Fig. 1).

In the process of converting natural images to Mooney images, information is lost. For instance, when examining a specific edge within a Mooney image, it is challenging to perceive whether the observed edge is a consequence of variations in depth that produced a shadow, alterations in illuminance, or modifications in coloration within the template scene (*Cavanagh, 2011*; *Hegdé & Kersten, 2010*; *Moore & Cavanagh, 1998*). Therefore, arriving at a "correct" understanding of the image depends on the observer's ability to resolve these ambiguities and interpret the black and white patches meaningfully. Consistent with a top-down modulation of perception, it has been shown that prior knowledge of the images' content significantly enhances the interpretability of Mooney images (*Dolan et al., 1997*; *Hegdé & Kersten, 2010*; *Milne et al., 2022*; *Teufel, Dakin & Fletcher, 2018*; *Teufel & Nanay, 2017*). Mooney images have also been used to investigate how humans perceive and mentally complete fragmented or incomplete visual stimuli (*Grützner et al., 2010*; *Mooney, 1957*; *Verhallen & Mollon, 2016*), the effect of top-down

processes on sensitivity of early visual processing (*Teufel, Dakin & Fletcher, 2018*), object recognition (*Dolan et al., 1997*; *Imamoglu et al., 2012*), face perception (*Latinus & Taylor, 2005*) and the neurophysiology of the visual system (*Hegdé & Kersten, 2010*; *Hsieh, Vul & Kanwisher, 2010*). There are even potential practical use cases; (*Castelluccia et al., 2017*) used Mooney images to create an innovative authentication method to replace classical passwords.

However, Mooney images have one major disadvantage: they are manually created in almost all previous studies. In Mooney's original study (*Mooney, 1957*), two-tone stimuli were created by hand-coloring the original template images.

While more recent studies apply thresholds to the digital image, the process is nevertheless both subjective and time-consuming. Typically, template images are first smoothed to remove noise and subsequently thresholded (*Schwiedrzik, Melloni & Schurger, 2018*). Smoothing is used to reduce noise in images, which is usually more noticeable in Mooney images than in template images. We think that the level of smoothing might affect how well Mooney images can be interpreted because too much smoothing could distort the edges, leading to less perceived closure and fewer illusory contours. Normally, these principles help to fill in missing edges and complete contours in Mooney images, making it possible to recognize objects in Mooney images (*Teufel, Dakin & Fletcher, 2018*). Thresholding is a procedure where each pixel's intensity is compared to a preselected threshold and subsequently set to either black or white (see Image thresholding techniques for more details). Both operations, smoothing and thresholding, can be implemented and performed easily using computers. However, the degree of smoothing and the threshold value still have to be determined manually or with some specific procedure to yield desirable Mooney images. This requires an iterative process (as described for example by (*Teufel, Dakin & Fletcher, 2018*)) which lacks well defined criteria due to its subjectivity. Consequently, the quality of the resulting Mooney images may vary across different laboratories and studies (*Nobis & Hunziker, 2005*).

To avoid these disadvantages and to be able to easily create large collections of Mooney images, it is necessary to create a technique to generate Mooney images automatically. A first attempt has been made by *Ke, Yu & Whitney (2017)* using a deep neural network. Their approach however is limited to Mooney images of faces. Furthermore, their network can only select a limited number of possible thresholds. The goal of the current study is therefore to investigate the perceptual effectiveness of more general techniques to create Mooney images.

To compare possible algorithms for Mooney image generation however, we must first define what an optimal Mooney image is. According to *Teufel, Dakin & Fletcher (2018*, p.8) for example "ideal two-tone images should be (i) experienced as meaningless black-and-white patches prior to having seen the template photograph. However, once participants have seen the template, they should (ii) give the strong experience of a coherent percept." This is a useful definition for most of the cases we have outlined above, in which the motivation is to dissociate low-level content from high-level understanding. However, there are different usages imaginable for which it might be necessary to create Mooney images which are quite easy to interpret. Examples are comparing human processing of

Mooney images to the processing of computer vision algorithms (*Zeman, Leers & de Beeck, 2022*) or chimpanzees (*Taubert & Parr, 2012*), where one might first display Mooney images which are easy to interpret for humans to check whether other species or algorithms are able to interpret them at all.

Because no single definition of an "optimal" Mooney image exists, we take an empirical approach to assessing a diverse set of algorithms. Our only objective was to assess and compare the extent to which humans could effectively interpret the Mooney images produced by these different algorithms. Portions of this text were previously published as part of a preprint (*Reining & Wallis, 2024*).

## Image thresholding techniques

We focus here on automatic thresholding techniques that use a global threshold, which are until now rarely used to create Mooney images (*Castelluccia et al., 2017*; *Imamoglu et al., 2012*; *Imamoglu, Koch & Haynes, 2013*), but are popular in image processing applications for applications such as image segmentation (*Chaubey, 2016*; *Lee, Yoon Chung & Park, 1990*; *Sezgin & Sankur, 2004*). It is of course possible that an algorithm's proficiency in tasks like image segmentation does not necessarily align with its ability to create Mooney images, as the algorithms were not designed for this purpose. This is what we want to experimentally assess in this article.

Thresholding is a procedure where each pixel's intensity in an image is compared to a predetermined threshold value. Pixels exceeding the threshold are assigned a value of 1 or 255 (depending on the color mapping), representing white, while pixels falling below the threshold are assigned a value of zero, representing black.

While thresholding is used in many areas of computer vision, one of the original applications of image thresholding is image segmentation into foreground and background. This process assumes that in general the pixels of foreground objects have a higher intensity than pixels being part of the background. By selecting a threshold lower than all object pixel intensities and higher than all background pixel intensities it would then be possible to perfectly separate objects from background. But of course this is not always possible, as in most pictures the intensities of foreground and background overlap. Nevertheless, many image thresholding algorithms were extensively tested with regards to their ability to segment images. Even though results are not perfect, there are some algorithms which perform well for specific kinds of images (*Sezgin & Sankur, 2004*). While for most applications of thresholding there exist a number of more complicated machine learning algorithms to perform the same task, simple thresholding algorithms remain appealing because, unlike *e.g.*, deep neural network approaches (*Minaee et al., 2022*), thresholding algorithms are relatively simple and transparent with respect to the features of the image that are used for processing. Over the years many automatic thresholding algorithms were created which use different features for threshold determination.

*Sezgin & Sankur (2004)* started to group thresholding algorithms with respect to the kind of features they use. Their work is one of the most influential works comparing and categorizing thresholding techniques. Even though new techniques arose in the past two

decades, the categories described by them are still widely used to describe thresholding algorithms (*Chaubey, 2016*).

In the following we want to give a brief overview over the different categories of thresholding algorithms, discuss which of them were of interest for us, present prominent examples and discuss their performance. Even though many performance measures are possible, in this context we will refer to the performance in background-foreground segmentation as it is the most evaluated and reviewed measure. However, we assume that algorithms performance in segmenting images does not *per se* correlate with performance in creating Mooney images which are easy or hard, respectively, to interpret. This is because a good segmentation algorithm would for example remove all inner contours in an object. These on the other hand might be important to recognize the object in a Mooney image as these contours are important for human object recognition in general (*Biederman, 1985*, *1987*).

The first major division between thresholding algorithms can be drawn between non-spatial and spatial techniques. Non-spatial techniques are simpler, because they only rely on the intensity values of the pixels without respect to their position and context in the image. Examples of non-spatial techniques include a method to threshold at the deepest concavities of the intensity histogram (*Rosenfeld & De La Torre, 1983*), minimizing expected misclassification error (*Kittler & Illingworth, 1986*; *Cho, Haralick & Yi, 1989*), maximizing information (entropy) between the pixels on either side of the threshold (*Kapur, Sahoo & Wong, 1985*), and clustering image pixels into two groups according to intensity (*Kittler & Illingworth, 1985*). The most popular non-spatial technique is Otsu's method (*Otsu, 1979*) which selects a threshold by maximizing the variance between two groups of pixels. Though other non-spatial methods can perform better in segmentation, they are also not as simple to use and make more assumptions.

Spatial image thresholding techniques are in most cases more complex. They take the location of a pixel in the image and the intensities of its surrounding pixels into account. This allows for the computation of higher-level features such as edges, pixel intensity correlations or spatial entropy. One can then choose to either select a threshold which maximizes the feature itself or the similarity between the features of the thresholded and the same features of the template image. Examples for these are the maximization of edge information in the thresholded image (*e.g.*, (*Dyke-Lewis, Weeks & Myler, 1993*; *Weeks, Myler & Lewis, 1993*; *Weeks, Myler & Apley, 1994*)) or maximization of the similarity between the edges of the thresholded image and the template image (*Belkasim, Ghazal & Basir, 2000*; *Samopa & Asano, 2009*). Most of the techniques in this category are newer ones and are often superior to non-spatial techniques (*Belkasim, Ghazal & Basir, 2000*; *Samopa & Asano, 2009*).

Besides the division into spatial and non-spatial categories, thresholding algorithms can also be applied globally or locally. Global algorithms are applied to the whole image at once (usually with a single threshold), whereas in local algorithms the image is divided into regions and for each region or sometimes even each pixel a separate threshold is determined based on local statistics (*Sezgin & Sankur, 2004*). For specific images with inhomogeneous lighting and especially for the case of document binarization this can be

advantageous. For the thresholding of natural images however, local methods tend to be worse than global methods techniques (at least with respect to segmentation performance (*Sezgin & Sankur, 2004*)).

We aimed to determine how the application of different threshold selection methods (and subsequently smoothing kernel sizes) affects the interpretability of Mooney images. As Mooney images were previously always generated with global thresholds and the fact that we could only compare four different techniques due to experimental resources, we decided to only compare global thresholding techniques in this study.

We generally expected spatial thresholding techniques to have a higher impact on interpretability of Mooney images since human vision itself is context dependent (though note that this study should be considered exploratory). To test this, we chose two non-spatial (mean threshold and Otsu) and and two spatial (max edge and edge similarity) methods, which are described in greater detail below. In our first experiment we experimentally compare these four algorithms, and in the second experiment we examine how the choice of the smoothing kernel size affects the interpretability of Mooney images created by the different algorithms.

# GENERAL METHODS

## Dataset

We used images from the THINGS dataset (*Hebart et al., 2019*; https://things-initiative. org/) due to the wide range of different concepts that it contains. These concepts are not only everyday objects but also more specific objects and animals. This wide variety of concepts was important to us as we wanted to produce as generalizable results as possible. Another reason for choosing the THINGS dataset are its labels and extensive annotations. Each of the more than 26,000 images belongs to one of the 1,854 concept groups labelled with concrete, picturable object names of everyday language. Furthermore, for a subset of the full image dataset there exists for one image per concept an embedding in a 49 dimensional space created by *Hebart et al. (2020)* using human similarity judgments. These 49 dimensions are "highly reproducible and meaningful object dimensions that reflect various conceptual and perceptual properties of those objects" (p. 1173 *Hebart et al., 2020*). They were useful to us as we used some of them for grouping similar concepts for foil answer generation in the second task in both of our two experiments.

The images taken from the THINGS dataset we used as templates in our study had to meet the following criteria. First, to ensure consistent image sizes and avoid the need for resizing, we opted for a standardized resolution of $800 \times 800$ pixels. Any images with different resolutions were excluded from the potential stimulus pool. Second, we removed all images with fewer than three images of the same concept. This was because the third task in the first experiment required multiple equally-sized images of the same concept. Finally, to be able to use the similarity embedding, we used only those images with existing embedding data. After applying all these constraints we were left with 549 images belonging to 549 different concepts. These images which we will from now on refer to as template images were then converted to Mooney images as described in the stimuli sections of experiment 1 and 2. One example image can be seen in Fig. 1.

### Threshold selection techniques

As described in the introduction, we selected four threshold techniques for this study, which are representative of various types of global thresholding techniques. These techniques are explained in more detail below.

#### Mean threshold

This threshold selection technique can be viewed as a simple baseline. It uses the mean intensity of all pixels as threshold. In a study by *Glasbey (1993)* it was found that the performance of this threshold selection method is not very high when it comes to background-foreground segmentation. Nevertheless, we included this technique for three reasons. The first one is its simplicity. Being this simple it is a technique everyone can use and understand without requiring much background knowledge. The second reason is that we are interested in techniques to create unambiguous Mooney images as well as techniques to create Mooney images which are very hard to interpret. And finally, as already stated in the Image thresholding techniques section, the ability of a algorithm to segment images does not have to influence how Mooney images generated by it are perceived.

#### Otsu's threshold

Otsu's threshold selection (*Otsu, 1979*) belongs to the clustering-based non-spatial threshold selection techniques. For it to work well the images it is applied to have to fulfill some constrains with respect to the histogram shape (*Bangare et al., 2015*; *Kittler & Illingworth, 1985*; *Lee, Yoon Chung & Park, 1990*). However, it still works better compared to more informed methods (*Sezgin & Sankur, 2004*). Furthermore, Otsu's method is one of the most used and well known threshold selection techniques. It is interesting for our study as it uses more information than the mean threshold selection method but still works on intensity values only and is therefore a non-spatial method.

It selects an intensity as threshold value that minimizes the intra-class variance of the black and white class. This is equivalent to maximizing the inter class variance and can therefore be viewed as a form of Fisher's Discriminant Analysis described by *Fisher (1936)*. We relied on the implementation of the scikit-image library.

#### Max edge threshold

We furthermore wanted to take techniques into comparison which also consider spatial features and not only an intensity histogram. As edges are important for human visual perception and detection of objects (*e.g.*, (*Biederman, 1985*)) we decided to implement two algorithms that work with edges as features.

The algorithm described here maximizes the number of edges in the thresholded image. This could lead to an ambivalent effect on the perception of the images content. On the one hand, more edges should provide more information to observers about the images content (*Attneave, 1954*; *Biederman, 1985*). Too many edges on the other hand, especially based on noise in the template image could also be useless for object recognition and make it harder by cluttering the scene instead (*Hegdé & Kersten, 2010*).

Thresholding algorithms which try to maximize edge information were already studied by *Dyke-Lewis, Weeks & Myler (1993)*, *Weeks, Myler & Lewis (1993)*, *Weeks, Myler & Apley (1994)*. We, however, for reasons of simplicity decided not to maximize edge information but total amount of pixels classified as edge instead. For this we iterated over all possible thresholds, applied them to the smoothed image and created an edge map of the thresholded image using an Canny edge detector (with $\sigma = 1$).

Canny edge detectors (*Canny, 1986*) are based on a Gaussian filter and are known to produce robust edge maps for a wide variety of images (*Maini & Aggarwal, 2009*; *Samopa & Asano, 2009*). The edge maps created by this edge detector are images with the same resolution as the template image but with all pixels being black. Only pixels which were classified as edge pixels are white. In the following we can simply count the number of pixels defined as belonging to an edge and select the threshold which produces the highest number of edge pixels.

### Edge similarity threshold

Like the previously described technique, this threshold selection technique uses edges to determine a threshold. This technique, however, does not simply try to maximize the number of edges. Its goal is rather to create a Mooney image which edges are as close to the edges of the template (smoothed) image. This seemed like a reasonable approach as we wanted to avoid unnecessary edges which are not present in the template image but rather an artifact of still existing noise in the template image.

Our implementation is inspired by *Belkasim, Ghazal & Basir (2000)*, *Samopa & Asano (2009)*, which both used slightly different methods and different similarity metrics but both achieved good results. Despite that, we decided on yet another similarity metric to compare the edge maps. That was because the metrics used by them are very sensitive to even small differences, which in our case arose due to the smoothing of our images. Thus, the metrics used by them produced unreasonable thresholds as results for the smoothed images.

The technique iterates over all possible thresholds while computing an edge map of the Mooney image for each possible threshold. Afterwards, it compares the edge map of the template image to the edge maps of the thresholded images by computing the Hausdorff distance between each template image and Mooney image. It then selects the threshold for which the distance between the two images is minimal. Therefore, the thresholded image should be as similar as possible to the template image in terms of edges.

The Hausdorff distance is a common metric for comparing images and is particularly well suited for comparing edge maps because of its tolerance of small positional errors (*Huttenlocher, Klanderman & Rucklidge, 1993*). In our application the Hausdorff distance can be described as the maximum of the shortest distances between each edge pixel in the template image and an edge pixel in the thresholded image. For our study we relied on the scikit-image implementation of the Hausdorff distance.

## Stimuli

In both of our experiments we showed Mooney images to the participants. Those Mooney images were created in a two step process, which can be seen in Fig. 1. First, we converted

the images to grayscale using the CIE1931 linear intensity mapping and applied a slight blur to all images, using a Gaussian kernel with a standard deviation that varied between the different experiments. For all values, the kernel truncated after four times the standard deviation. Afterwards, all images were mapped to an 8-bit grayscale format to get meaningful and comparable thresholds after thresholding.

In the second step we computed four thresholds for each image using the four different threshold selection methods described above. After threshold selection we converted the smoothed grayscale images to Mooney images by applying the computed thresholds.

The resolution of 800 × 800 pixels and the configuration of the experimental setup resulted in a stimulus size of 10.5° × 10.5° of visual angle. This size was on the horizontal axis quite similar to the size of the Mooney images used by *Teufel, Dakin & Fletcher (2018)*. Vertically, however, our images were larger due to the square aspect ratio.

One of the participants' tasks was to identify the concept shown in the image from a row of four concepts (see Procedure below). To make this task harder, the foil concepts were chosen from concepts that have similar shapes to the true concept. To determine concepts with a similar shape, we used the embedding provided by *Hebart et al. (2020)*. From the 49 dimensions provided by the embedding we selected 13 (see "Foil Answers for Concept Detection Task") which were shape related and projected all concepts with existing embedding data in this new shape related space. We used this new space to select the three concepts for each of our images which were closest to the image's true concept in terms of Euclidean distance in this space. As can be seen in the example in Table A1 the foil answers selected like this are indeed more related to the correct answer than randomly selected foils. For each presentation of Mooney images generated from templates with this object category we reused the three foils concepts. The order of the four alternatives was shuffled each trial.

## Equipment

The code for the experiment in stimulus creation was written and executed in Python 3.10.6. For the latter we furthermore used the scikit-image (0.20.0), pandas (2.0.0) and the scipy (1.10.1) library.

The stimuli were presented on an LG UltraGear 27GN950 monitor connected to a computer running Ubuntu 20.04. The monitor had a spatial resolution of 3,840 × 2,160 and a temporal resolution of 144 Hz. For gamma correction a X-Rite i1DisplayPro colorimeter was used. As the displayed stimuli did not contain any colors no precise color correction was necessary.

A chin rest was used to fix the observers' distance to the monitor at a distance of 55 cm. This resulted in the monitor covering about 57 degrees of visual angle. All experiments took place in a darkened laboratory of the AG Perception at TU Darmstadt.

## Ethics

The procedures adhered to the Standard 8 of the American Psychological Association's "Ethical Principles of Psychologists and Code of Conduct"(2010) and were approved by

the Technical University of Darmstadt Ethics Commission (Application number EK 77/2022).

## Data analysis

For general evaluations and visualization of the data analysis Python (v3.10.6) with the packages numpy (v1.23.5), pandas (v2.0.0), matplotlib (v3.7.1) and seaborn (v0.12.2) was used. Statistical tests were conducted using the pingouin (v0.5.3) Python package. The specific process for the analysis of the data of our two experiments can be seen in the respective sections.

# EXPERIMENT 1

## Methods

For this experiment we used a single-subject design and considered each participant as a replication (*Smith & Little, 2018*).

### Participants

One female and one male observer (20 and 21 years old, respectively) took part in the experiment. Both had corrected-to-normal vision. While none of them had seen neither the Mooney images nor the template images before, both participants were familiar with the research question. One of the participants was the author of the current study. The stimulus generation pipeline was tested on separate images to the main experiment, so the author had never seen the test images nor the templates before participating. The other participant was a TU Darmstadt student who was rewarded with 15 € per hour. Both participants provided written informed consent.

### Stimuli

The stimuli were Mooney images generated from a subset of images of the THINGS dataset. This subset contained 549 images which were chosen with respect to the constrains described in the general methods dataset section. We randomly selected 500 of these as a basis for the stimuli used in our experiment. The remaining 49 were used as basis for the stimuli in the practice trials.

All images were converted to four different Mooney images in a two step process which can be seen in Fig. 1 and which is described above in the general methods section. In this experiment we set the standard deviation of the Gaussian filter kernel to 2 px. We chose this value as it was the minimal kernel size that removed just enough noise in order to achieve the characteristic Mooney-image look. This characteristic entails the presence of numerous black and white patches within the image which are sufficiently large enough to not be the effect of noise in the template image.

### Procedure and design

In this experiment we investigated four different conditions, each one corresponding to one thresholding method. It is known that the quality of thresholding algorithms for object segmentation strongly depends on the images they are applied to *Sezgin & Sankur (2004)*. Therefore, we reasoned this could also hold for the ability to create Mooney images. As a result, we presented each template image for all different conditions. Consequently, over
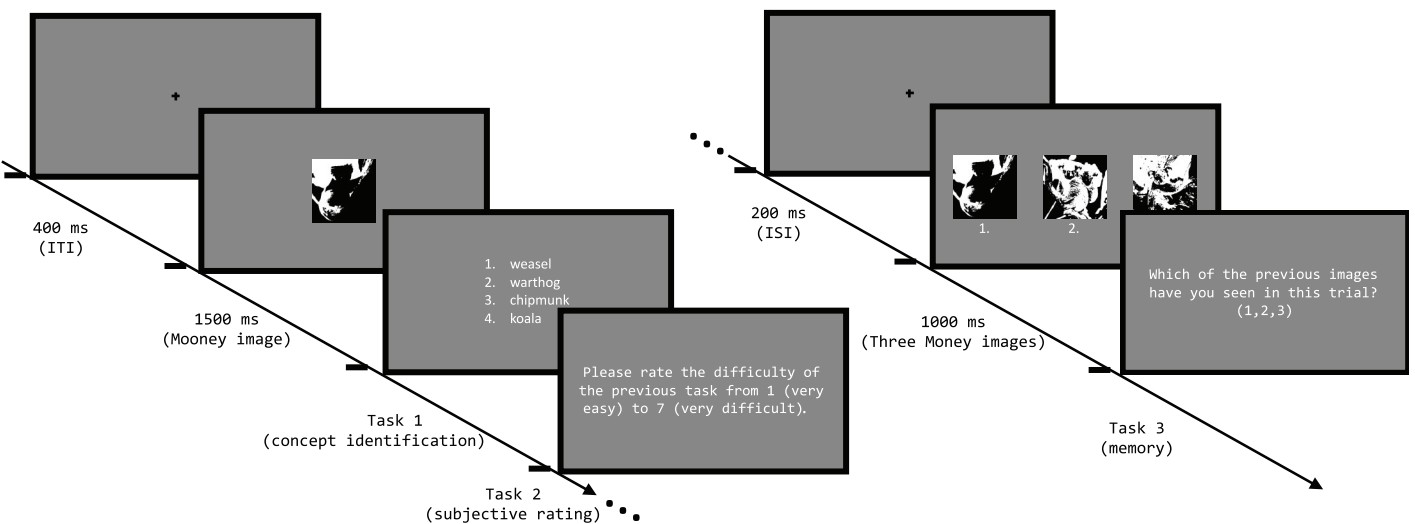

**Figure 2 Procedure of one trial of experiment 1.** Each trial consisted of three tasks. In the first task (identification task) the participants were presented with a Mooney image and four different concepts. The participants were supposed to indicate which of these concepts they saw in the Mooney image. In the second task participants were asked to give a subjective rating about the difficulty of the previous task. In the final third task (memory task) three different images were presented to the participants. The participants task was to detect which of these images was shown previously in this trial. The Mooney image is a modified version of one of the images taken from the THINGS dataset (*Hebart et al., 2019*), which is shared under a CC-BY license. It was modified according to the methods section. 

the course of the experiment each of the original images was presented four times—each of those presentations as a slightly different Mooney image. Applying four different conditions to 500 image and showing each image exactly once per condition resulted in a total of 2,000 trials per participant.

The main goal of this experiment was to investigate the ability of threshold selection techniques to create ambiguous or respectively unambiguous Mooney images. To gain information about this effect three tasks (two based on performance and one based on a subjective rating) directly followed the presentation of a Mooney image. They differed from the tasks used in previous experiments with Mooney images (*e.g.*, (*Teufel, Dakin & Fletcher, 2018*)) because we did not only want to test whether certain edges or presence of objects were detected or not but rather if the participants were able to detect the content of the image.

Each trial started with the presentation of a fixation spot for 400 ms to center the gaze of the participants Fig. 2. Afterwards a Mooney image was shown for 1,500 ms. This is the same timing as used by *Teufel, Dakin & Fletcher (2018)* to allow for the visual system to interpret the black and white spots of the Mooney image as a meaningful concept if possible. Immediately afterwards the first task started. This first task was a four-alternative identification task and will from now on be referred to as the identification task. Four different but shape related concepts were presented to the participants as words on the screen. The task was to identify which concept was shown in the previous image. This allowed us to measure if the participants were able to recognize the image's content in a task with an objectively correct answer. In a pilot study we used an open (free) response design and found that performance was very poor, so we used this four-alternative task here.

After the participants gave their answer by pressing a key on a keyboard, the second task directly appeared on the screen. Here, the participants were asked to give an additional subjective rating of the difficulty of the previous task on a scale from one (*very easy*) to seven (*very difficult*). Again, the participants were supposed to indicate their rating by pressing the corresponding key.

This triggered the beginning of the final task (memory task), which started with the presentation of a fixation spot for 200 ms. Subsequently, three different Mooney images all showing the same concept and thresholded with the same thresholding method appeared on the screen for 1,000 ms. Afterwards participants were supposed to indicate which of the images they had previously seen in this trial. This task was added to our experiment as yet unpublished results by WJ Harrison (2023, personal communication) indicate that recognizing the object in a Mooney image increases the probability to recognize it between unseen images. Once more, the answer was given by pressing the corresponding key. This triggered the next trial starting with the presentation of the fixation spot.

In all of the described tasks the participants had as much time as they needed for giving the answers. To reduce possible learning effects, no feedback was given in any of the tasks.

The 2,000 trials were randomly split into 40 blocks of 50 trials. The 40 blocks were divided into three sessions of 100 min each on consecutive days. While sessions two and three consisted of 14 blocks each, the first session consisted of only 12 blocks. At the beginning of the first session, however, participants had to start with one practice block to get familiar with the task. The training block consisted of only 25 trials, with stimuli displayed twice as long as in the first ten trials. The stimuli in the practice trials were taken from the practice stimuli pool and the results were not included in the analysis. After finishing the practice trials, participants were only allowed to start the real experiment if they got at least 15 correct answers in the identification and memory task. Otherwise, they had to repeat the practice trials until the criterion was met.

### Data analysis

First, we compared the results of the different threshold selection techniques independently of the behavioral data. The first thing we were interested in was whether some threshold selection techniques tend to select higher or lower thresholds than others. Therefore, we computed boxplots for the distributions of all thresholds selected by one threshold selection techniques over all of the 500 images. This not only allowed us to compare the general tendencies of each algorithm, but also to find out whether some selected more variable thresholds than others. Furthermore, for each image we computed the standard deviation of the four thresholds selected by the different thresholding techniques. Computing the mean of the standard deviations over all images allowed us to judge whether the algorithms performed indeed differently as intended. To capture the degree of variation in the resulting images, we computed pairwise dissimilarity ratios between the Mooney images generated by different algorithms but originating from the same image. The dissimilarity ratio between two Mooney images is the ratio of the number of pixels which are white in one of the images but black in the other or *vice versa*. A value of one would mean that the Mooney images are polarity-reversed versions of each other

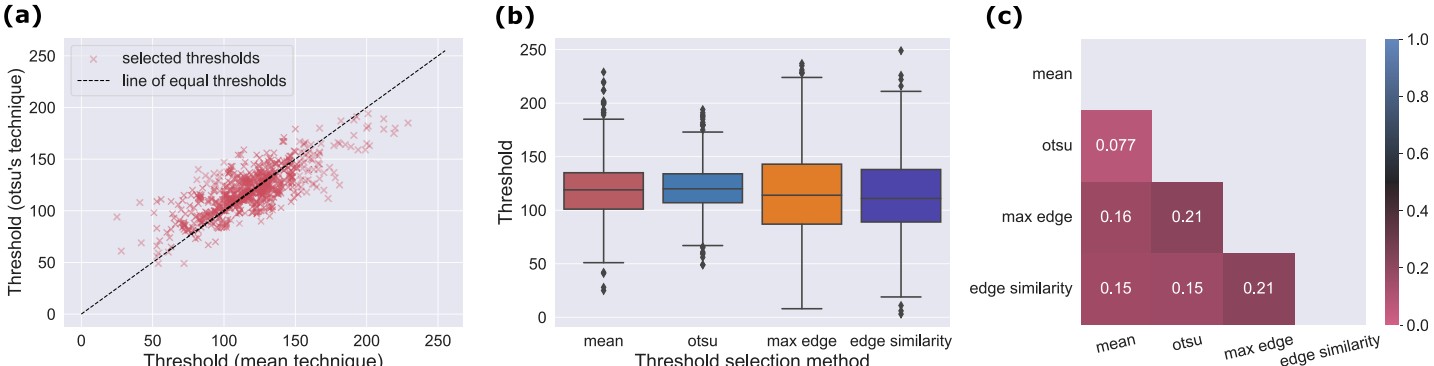

**Figure 3** **On average all techniques select similar thresholds but produce very different Mooney images.** (A) Comparison of threshold values selected over all images. Boxplots describing the threshold distributions over all images for the different thresholding techniques. (B) Direct threshold comparison between two selected thresholding techniques Shown are the threshold selected by the mean and the Otsu technique. While the points indicate the thresholds selected for each image, the solid line indicates where the points would lie if both techniques selected the same thresholds. (C) Mean dissimilarity ratios between pairwise combinations of thresholding techniques over all images. The dissimilarity ratio describes the ratio of Mooney images' pixels as generated by two different technique which is white (or black) in one image but black (or white) in another.

(every white pixel in one image is black in the other), whereas a value of zero would mean the images were identical. (Such images would of course be in some sense quite similar to each other but such a complete flip of polarities is not possible as we only used global thresholding techniques which maintain the ordering of the pixel intensities).

Second, we examined the behavioral data. We computed the overall performance of the three tasks as measured by proportion of correct answers (proportion correct) for the first and third task and the mean of the subjective rating for the second task. This allowed us to compare the results between the two participants as well as between the different conditions. Uncertainties of the estimates of proportion correct and the mean of the rating were always quantified by giving the 95% confidence interval, which was computed by multiplying the standard error of the mean with 1.96. Additionally, for each participant and task separate one-way analysis of variances (ANOVA) were conducted to investigate the significance of the effects of the different thresholding techniques.

## Results

### All techniques selected similar thresholds on average but produced different Mooney images.

On average the thresholds selected by the four thresholding techniques were quite similar. As shown in Fig. 3A the distribution of the selected thresholds for all images are approximately centered around the templates images' mean intensity (119). Minor differences like the mean thresholds of the max edge and edge similarity technique are not important with respect to the high variance of the thresholds. The threshold variances however, varies between the different techniques. It is noteworthy that the standard deviations of the two spatial thresholding techniques are higher ($\sigma_{max\,edge} = 44.39$, $\sigma_{edge\,similarity} = 36.69$) than the standard deviations of the non-spatial techniques ($\sigma_{mean} = 30.68$, $\sigma_{otsu} = 23.60$) (see Fig. 3A).
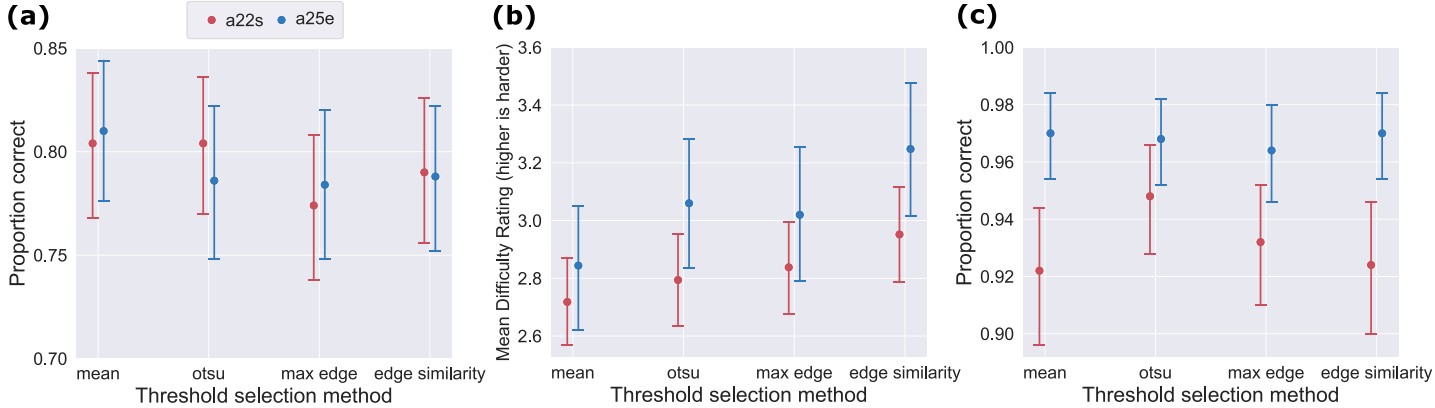

**Figure 4 Only difficulty rating produced slightly different results for different thresholding techniques.** (A) Proportion correct in the identification task. Depicted are the proportion corrects with 95% confidence intervals (vertical lines) for the two participants and the four different threshold selection techniques. (B) Subjective difficulty ratings. Depicted is the mean and the 95% confidence interval of the subjective ratings of each participant for each thresholding condition. A higher rating indicated a higher difficulty and *vice versa*. (C) Proportion correct in the memory task. Depicted are the proportion corrects with 95% confidence intervals for the two participants and the four different threshold selection techniques.

However, even though the thresholds were quite similar in the mean, on the image level the threshold selected by the different techniques varied substantially. As an example, Fig. 3B shows a scatter plot of mean and Otsu thresholds. This is also supported by the fact that the mean standard deviation between the thresholds determined for each image was 20.76.

The differences between the thresholding techniques are more evident when looking at the mean dissimilarity ratios in Fig. 3C. For the techniques which produced the Mooney images with the biggest differences, more than one fifth of the pixels was different depending on the technique. Even for pairwise comparisons with less difference, more than 15% of the pixels varied for most combinations in the mean.

## Performance in the identification task did not differ significantly for different thresholding techniques

The performance as measured by proportion correct in identifying the correct concept in the presented Mooney image was very similar for all thresholding techniques. This can be seen in Fig. 4A. All proportion correct values lie between 0.77 and 0.81. This interval is rather small considering the fact that all proportion correct values lie in the 95% confidence interval of each other. The one-way ANOVAs for each participant both indicated non-significant results, $F(3, 1{,}996) = 0.62$, $p = 0.6$ for participant a22s and $F(3, 1{,}996) = 0.44$, $p = 0.721$ for a25e.

The performance was also similar between the participants. Both had a mean proportion correct of 0.79 and similar variations in their performance. Furthermore, for thresholding techniques in which the performance of one participant was slightly increased, the performance of the other was also increased. The same holds for conditions in which performance was slightly lower.

**Subjective difficulty ratings differed only slightly for different thresholding techniques.**

Participants' subjective difficulty ratings of the identification task varied only slightly over techniques. In Fig. 4B we can see that the mean difficulty rating varied with respect to the 95% confidence interval more than the proportion correct seen in Fig. 4A. Even though most means still lie in the 95% confidence interval of other, the total differences are greater. At least for the conditions with the lowest (easiest reported difficulty) (mean threshold selection) and highest rating (hardest reported difficulty) (edge similarity threshold selection) the means were not in each others confidence interval. Nevertheless, the confidence intervals still overlap. The two remaining conditions have quite similar difficulty ratings which lie beneath the ones for the mean and max edge technique. The one-way ANOVAs for each participant again both indicated non-significant results ($F(3, 1,996) = 1.37$, $p = 0.249$ for participant a22s and $F(3, 1,996) = 2.07$, $p = 0.102$ for a25e). In general, the ratings of participant a25e were slightly higher than those of a22s, but the general trends described above can be found in either participants data.

## Performance in the memory task did not differ significantly for different thresholding techniques

As for the previous tasks results did not vary much between the different thresholding conditions. In Fig. 4C we can see that for participant a25e all proportion corrects varied by less than 0.01 and are in each others confidence intervals. The proportion corrects for participant a22s are more scattered. While the values for the mean, edge similarity and max edge technique also only vary by less than 0.01, the performance for the Otsu's threshold selection technique is slightly increased. This does not reproduce for participant a25e. Again, both ANOVAs report non-significant results, $F(3, 1,996) = 1.09$, $p = 0.35$ for participant a22s and $F(3, 1,996) = 0.13$, $p = 0.942$ for a25e.

In general, participant a25e was slightly better (proportion correct = 0.97) than participant a22s (proportion correct = 0.93) achieving nearly perfect results.

## Discussion

In this experiment we compared four different thresholding techniques with regards to the Mooney images they created and the interpretabilty of those images as measured by two performance based tasks and one subjective rating. Overall, our results showed that the different thresholding techniques did not greatly influence the recognizability or memorability of Mooney images, even though the techniques did produce physically different images.

First, these results show that none of the techniques was biased to select consistently high or low thresholds and therefore selected only reasonable thresholds around the images' intensity mean. Extreme thresholds would create Mooney images which are impossible to interpret, as extreme thresholds would result in images with all black or all white pixels. However, there are still differences in variance between the different algorithms, with the variance of the two spatial techniques being greater. This indicates that those techniques were more flexible when selecting a threshold, which both could

potentially result in better or worse Mooney images depending on the quality of the technique.

Furthermore, by looking at the dissimilarity ratios and standard deviations between the four thresholds for each image, we found that even variations with a standard deviation of 20 in pixel intensity results in very different Mooney images. Especially the large difference between the Mooney images of two spatial techniques is noteworthy as this shows that Mooney images created by the max edge technique contain many edges which are not present in the template image.

The behavioural results show that even though the Mooney images vary, the interpretability as measured by the three tasks is almost identical for all thresholding techniques. The unimportance of the small differences of proportion correct in the identification task is supported by the results of the difficulty rating and the memory task. In all three tasks all algorithms not only achieved similar results, but there was also no systematic increase or decrease in performance and rating for one specific technique. This, however, goes with the caveat that proportion correct in the memory task was so high that possible differences in memorability might not have been visible in the data due to ceiling effects.

However, the fact that we did not measure differences in the responses for the different thresholding techniques may also arise from our experimental design. The performance in the identification task was unusually high for Mooney images, which are usually considered hard to recognize (*Cavanagh, 2011*; *Moore & Cavanagh, 1998*). We see three possible causes for this. First, the images of the THINGS might have been so representative for their specific concept that in many images the objects are shown on a plain background. A plain background without clutter on the other hand makes concept identification easier (*Hegdé & Kersten, 2010*). Therefore, the images of the THINGS dataset might become Mooney images which are easy to interpret in general. Second, it was reported by both participants that by seeing multiple images of the same concept in the memory task, they were sometimes able to recognize the correct concept from the additional images and remember it for future presentation of Mooney images originating from the same image. Therefore, it is possible that the identification task was not answered correctly because the actual identification of the concept was rather done using memory. This would also explain the indifferent subjective difficulty ratings as a task gets easier regardless of whether it is done by memory or perception. Third, using a text-based response for the identification task makes the possible concept(s) more clear than in a Mooney image; therefore people may have been able to improve their recognition after seeing the response options.

## EXPERIMENT 2

To mitigate some of these limitations, we altered the experimental design in Experiment 2. We removed the memory task from the experiment, changed the difficulty rating into a visibility rating and selected a subset of images which, on average, should be harder to interpret. In addition, as smoothing is another important step in Mooney image generation, we decided to also compare different smoothing kernel sizes in the second experiment to see whether the interpretability might be more dependant on the smoothing

than on the thresholding. Finally, here we also targeted the population average performance rather than examining individual participant performance, and so tested more participants with fewer trials per participant.

## Methods

### Participants

A total of 15 participants took part in the experiment. Nine of them were female and six male with a mean age of $22.2 \pm 2.97$ ($M \pm SD$). All of them were naive observers who were not familiar with the purpose of the study. They all had normal or corrected-to-normal vision and were students of TU Darmstadt. All participants certified their informed consent and were rewarded with either 15 € per session or one participation credit for a course assignment.

### Stimuli

The stimuli in this experiment were again Mooney images generated from a subset of images taken from the THINGS dataset. This subset consisted of 40 images and was yet another subset of the 500 images used in the first experiment. To increase the difficulty of the second experiment and to account for images of a range of difficulties, the 40 images were sampled as follows: The 500 images used in experiment 1 were divided into three groups. The first group consisted of "easy" images for which the correct concept was detected in more than 75% of the presentations as different Mooney images by the two participants in Experiment 1. The second group consisted of "medium" images for which concepts were detected correctly in more than 25% and up to 75% of the cases. The third group consisted of "difficult" images for which concepts were detected in less than or equal to 25% of the cases. We randomly selected 13 images of the "easy" group, 14 images of the medium group and 13 images of the "difficult" group, but filtered out all images for which concept nouns might not be known to new participants due to language difficulties.

The 40 images were again converted to Mooney images by the process described in the general methods section. For this experiment however, we did not only vary the thresholding technique in the process of generation, but also varied a second dimension. Before applying the four different threshold techniques one of the template images was converted to three different smoothed images using three different standard deviations of the gaussian smoothing kernel (two, four and six pixels). We selected these three specific values for the following reasons: First, a minimum standard deviation of two was necessary to effectively eliminate a major portion of the noise in the thresholded images. Second, for thresholds exceeding six, we observed that there were no longer any illusory contours visible to us. Thus, these chosen values strike a balance between noise reduction and the emergence of illusory contours, which both are characteristic properties of Mooney images (*Castelluccia et al., 2017*; *Teufel, Dakin & Fletcher, 2018*). Furthermore, the study by *Ke, Yu & Whitney (2017)*, which proposed a method for automatically generating Mooney faces, also used standard deviations between two and six. The effect of varying the size of the smoothing kernel on a Mooney image can be seen in Fig. 4A.

**(a)**       **(b)**

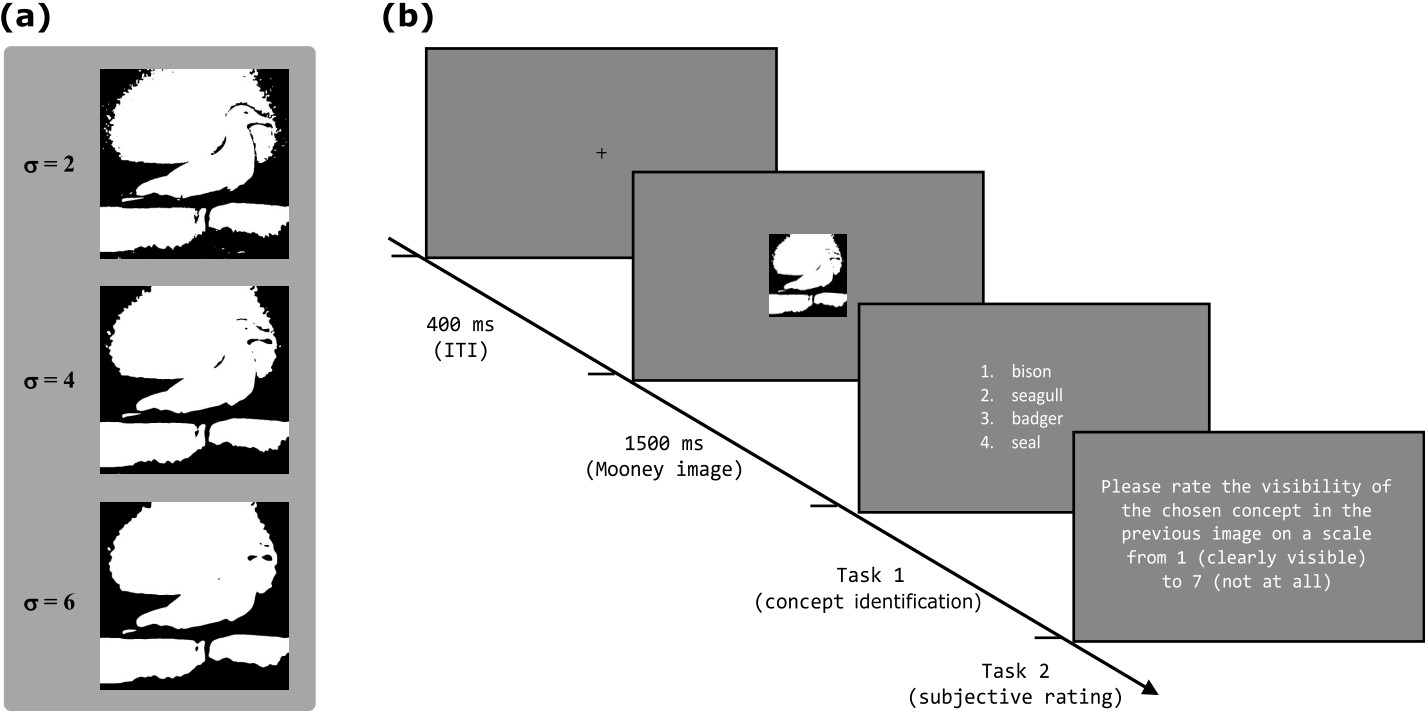

**Figure 5 Stimuli and Procedure for experiment 2 (A) Mooney images generated using different standard deviations of the gaussian filter kernel and the mean luminance as threshold. (B) Procedure of one trial of experiment 2. It is very similar to the procedure of experiment 1 as seen in Fig. 2.** The only differences are that in the second experiment the third task is missing and the text which asks for a subjective rating was slightly altered to ask for a visibility rating and not a difficulty rating. The shown image is a modified version of one of the images taken from the THINGS dataset (*Hebart et al., 2019*), which is shared under a CC-BY license. It was modified according to the methods section.

Applying four different thresholding algorithms to three differently smoothed images resulted in a total of twelve different Mooney images for each of the 40 template images. This made a total of 480 different Mooney images.

The stimuli for the practice trials were the same as in experiment 1.

### Procedure and design

The procedure of each trial was as in experiment 1 with two exceptions Fig. 5B. First, we omitted the third task (memory). It was originally meant as a secondary measure of how well the participants understood the presented Mooney images. Reports by the participants however suggested that observers might learn the content of the images by combining the information from the three pictures in the third task, which all showed the same concept. Therefore, to avoid such learning effects, we removed this task. Second, we changed the subjective rating. In the previous experiment we asked the participants for a difficulty rating of the task. This rating might also have been influenced by learning effects. To more precisely target participants' subjective impression of their understanding of the Mooney image, participants were asked to judge how well the concept they had chosen in the first task was visible on the Mooney image on a scale from one (*clearly visible*) to seven (*not at all*). We reasoned that even if participants would learn some concepts, they would still judge the visibility based on the quality of the Mooney image only.

As in the first experiment, we adhered to the principle of showing each image exactly once for each condition. Each of the selected 40 template images was presented twelve times over the course of the experiment-each time as a different Mooney image (four threshold selection techniques times three gaussian smoothing kernels), for a total of 480 trials per participant. The 480 trials were split into twelve blocks of 40 trials, with breaks in between. In each block we ran through all of the 40 template images but in a random order and with randomly changing conditions. This made sure that none of the original images were presented twice in the same block.

All twelve blocks were performed in one session of 60 min. As in the first experiment, one practice block with 25 trials, of which 15 needed to be correct, had to be performed by the participants before the beginning of the real experiment. One participant failed to get 15 out of 25 correct in the first practice block and had to repeat the practice.

### Data analysis

As in the first experiment, we computed the proportion correct in the identification task as well as the mean of the subjective ratings as measures of how well the images' content had been perceived in each of the twelve conditions. To measure the uncertainty of these measures we again computed the 95% confidence intervals using the standard error of the mean. Variance in subjective visibility ratings was quantified using a $4 \times 3$ repeated-measures ANOVA. The subjective visibility ratings were flipped to make data presentation more intuitive.

To quantify differences in proportion correct as a function of the experimental conditions, we estimated the posterior parameter distributions of a set of generalized linear (logistic) mixed models. These models allow us to capture inter-participant and image differences while estimating average tendencies and variance in the data.

We estimated the fixed effects of threshold selection method and size of the smoothing kernel as well as their interaction along with random effects for participant and template image on the probability to correctly identify the concept in an image. To evaluate whether kernel size and/or thresholding technique significantly influenced the probability of correctly identifying the concept we created three additional models. The second model was similar to the one described above but without the interaction term between thresholding technique and kernel size. Models 3 and 4 lacked either the factors for thresholding technique or kernel size. For descriptions of model parameters however, we will always refer to the first (full) model. The remaining models will only be used for comparison.

The parameters of all models were initialized with weakly informative priors, providing plausible scale estimates to the parameters without biasing the model towards any particular differences between conditions. The posterior distributions of the models were estimated using a Markov Chain Monte Carlo procedure. More details regarding the modelling can be found in "Generalized Linear Mixed Models". All models were implemented in the Stan language (v2.26.1 (*Stan Development Team, 2022*)) with the wrapper package brms (v2.19.0 (*Bürkner, 2018*)) in the R statistical environment (v4.3.1).
**(a)**

**(b)**

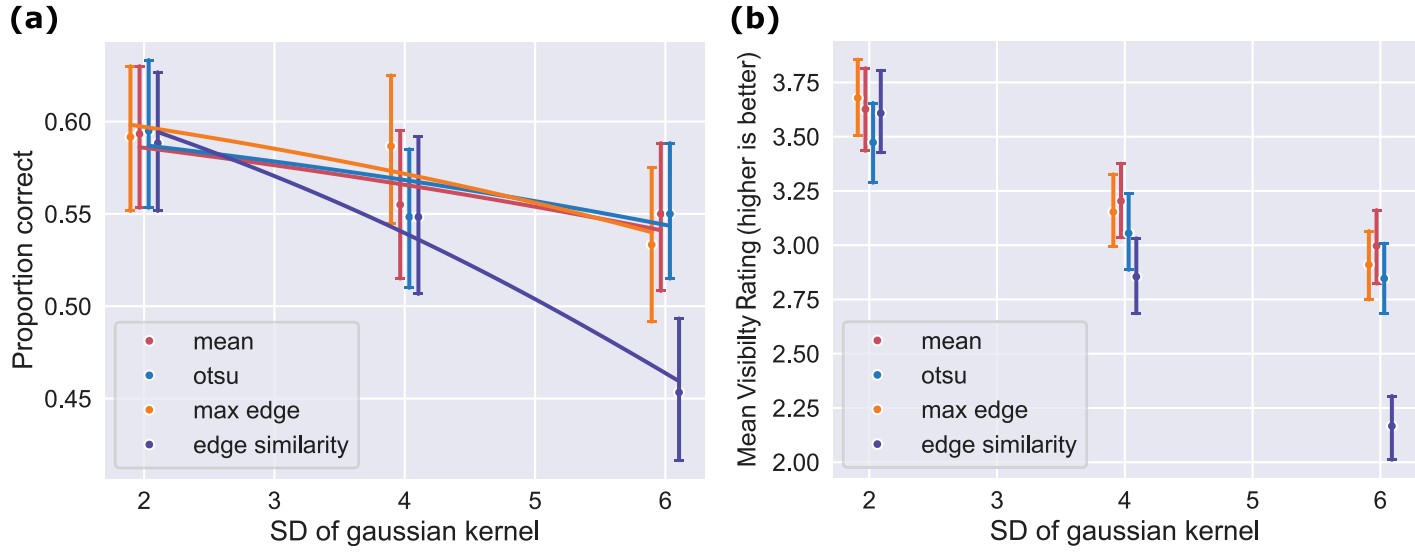

**Figure 6** **Results for all thresholding techniques and kernel sizes were similar except for the edge similarity technique.** (A) Proportion correct in the identification task. Depicted are the proportion corrects for the different thresholding techniques and kernel sizes. Errorbars indicate the 95% confidence intervals while the solid horizontal lines show the posterior mean from the mixed-effects model with the best fit. (B) Subjective visibility ratings. Depicted is the mean and the 95% confidence interval of the subjective ratings of each participant for each thresholding technique and kernel size. A higher rating indicated a better visibility and *vice versa*.

The different models were compared by their expected log-predictive density (ELPD), which is an estimate of how well the model generalizes to new data. This score was computed using the loo package (v2.6.0) which utilizes LOOIC as an approximation to the leave-one-out log likelihood (*Vehtari, Gelman & Gabry, 2017*). For model comparisons we will always present the difference in ELPD and standard error of the difference (SE).

### Results

**Thresholding techniques yielded mostly similar results while smoothing demonstrated significant impact on concept identification**

The results of proportion correct were very similar for all thresholding techniques. Only the edge similarity technique yielded a lower proportion correct for the most smooth blur kernel (Fig. 6). This is supported by a model comparison, which revealed an ELPD difference of 9.4 (with relatively large uncertainty of SE = 8.1) between the full model (containing both smoothing and thresholding technique and their interaction as predictors) and a model which did not take thresholding technique into account (only smoothing).

Smoothing on the other hand influenced the probability of correctly identifying concepts more. This main effect can not only be seen in Fig. 6 but is also supported by the following arguments. First, the mixed model without a representation of kernel size has a by 44 (SE = 10) lower ELPD than the full model. And second, the full model describes the posterior probability that the kernel size has a negative influence (as described by its coefficient $\beta$) on concept identification as $p\,(\beta < 0) > 0.999$ with $\beta = -0.15$ and 95% CI $= [-0.21$ to $-0.08]$.

As described above the results for the mean, otsu and max edge thresholding technique were very similar for all different smoothing kernel sizes. Only the edge similarity technique yielded less correct classifications for the highest smoothing condition. This, however, is not enough for the model comparisons to confirm a large interaction effect between smoothing kernel size and thresholding technique. This can be seen as the model with and the one without interaction received very similar ELPD scores (ELPD difference = 2.3, SE = 3.4).

### Subjective visibility ratings show a similar pattern to proportion correct

The findings of the mixed effects analysis of proportion correct in the concept identification task align with the subjective visibility ratings. Consistent with the fact that the full model performed best in the mixed effects analysis, the results of the two-way repeated measures ANOVA indicate a significant main effect on the visibility rating for thresholding technique, $F(3, 42) = 16.10$, $p < 0.001$, and a significant main effect for smoothing kernel size, $F(2, 28) = 25.43$, $p < 0.001$.

For the highest amount of smoothing the edge similarity technique not only received the lowest proportion correct but also got rated as producing the least visible images. This is reflected in the two-way repeated measures ANOVA which reported for the visibility rating a significant interaction between thresholding techniques and kernel size, $F(6, 84) = 9.02$, $p < 0.001$, even though the results for the remaining thresholding techniques were very similar.

### Probability of recognizing objects strongly depended on template image

Our mixed-effects model also revealed that the template image had a major influence on the probability of correctly interpreting the created Mooney images. The standard deviation of the distribution of image specific intercepts had a posterior mean of 2.25 (95% CI = [1.81, 2.80]). The same standard deviation for the participant specific intercepts was only 0.52 (95% CI = [0.33, 0.81]). This indicates that the variance in performance associated with different template images was approximately four times larger than the variance between different participants.

### Discussion

In this analysis we compared four different thresholding techniques as well as three different degrees of smoothing with regards to the interpretability of the generated Mooney images as measured by one performance based task and one subjective visibility rating.

Our results demonstrate that the recognition of objects in Mooney images is for all degrees of smoothing mostly independent from the four compared thresholding techniques. This aligns with the results from our first experiment, indicating that the similar performance of the thresholding techniques is not a result of the experimental design of the first experiment.

Only the edge similarity technique produces worse results for larger degrees of smoothing. This effect might arise as the smoothing distorts the edges of the original image. When using the max edge similarity technique to choose a threshold for smoothed template images and Mooney images, it is possible that the selected threshold could emphasize edges that are not crucial for recognizing objects in the original, unsmoothed image. In other words, the chosen threshold might effectively highlight irrelevant edges, potentially affecting the accuracy of object recognition.

On the other hand, the amount of smoothing represented by the size of the smoothing kernel significantly influenced the recognition of the shown objects. A high amount of smoothing significantly decreased the probability to recognize a depicted object. Likely, this is caused by the distortion of the edges produced during smoothing. First, the distortion of the objects' edges in the Mooney image makes it *per se* harder to detect. Second, the distortion of edges in the smoothed template could also lead to less closure and illusory contours perceived from the Mooney image. Normally, these principles are used to fill in missing edges and complete contours in Mooney images to enable object perception (*Teufel, Dakin & Fletcher, 2018*). However, if parts of existing contours and edges are already distorted and therefore misaligned the completion of missing parts of those contours as seen for example in Mooney images might be much harder (*Field, Hayes & Hess, 1993*). Furthermore, as smoothing reduces the number of edges and amount of details, less information is available to the observers to interpret the images. For these reasons we conclude, that smoothing indeed increases the difficulty to interpret Mooney images.

However, the choice of template images seems to have an even stronger influence. We showed that the probability of correctly identifying a depicted object varies a lot from one template image to another. These deviations were far bigger than those achieved by manipulating the size of the smoothing kernel or thresholding technique. Analyzing which image features are associated with resulting Mooney image interpretability variation would be a useful avenue for future work.

## SUMMARY AND CONCLUDING DISCUSSION

In two experiments we investigated the influence of smoothing and different thresholding techniques on the interpretability of Mooney images. We generated Mooney images from images of the THINGS dataset (*Hebart et al., 2019*) using different smoothing kernel sizes and thresholding techniques. These images were shown to participants and the interpretability was measured *via* performance-based multi-alternative identification tasks as well as subjective ratings.

Our results demonstrate that the recognition of objects in Mooney images is mostly independent from the four compared thresholding techniques. Moreover, it seems that as long as the exact threshold is reasonably chosen, somewhere around the image's mean, it does not significantly impact the interpretability of Mooney images in these images. This has potential implications on the generation of large Mooney image datasets in the future. In particular this means that one neither has to select the most suitable threshold in a tedious, manual procedure nor use complex algorithms for threshold selection as the

resulting Mooney images might be of similar recognizability anyway. There seems to be no single global threshold which creates the most ambiguous or most unambiguous Mooney image.

Nevertheless, depending on the goal of the studies using Mooney images it might still be necessary to generate Mooney images with varying interpretability. Our results show that this cannot be achieved by only using one of the compared thresholding algorithms. Rather, we suggest to manipulate the amount of smoothing which is applied before thresholding. By increasing the smoothing we were able to achieve a minor decrease in the probability of correctly interpreting a Mooney image (in particular for the edge similarity technique).

Additionally, the interpretability can also be manipulated by the choice to the template images. We showed that the probability of correctly interpreting a Mooney image varies greatly between different template images. For example, we speculate that Mooney images created from noisy and cluttered images are much harder to interpret than images with a uniform and noiseless background (*Hegdé & Kersten, 2010*).

But what if we want to increase or decrease the interpretability of Mooney images even more? After comparing global thresholding techniques from various categories as described by *Sezgin & Sankur (2004)*, we believe that global techniques are not flexible enough for this task. 8-bit grayscale images have only 256 possibilities to select a threshold. Although our study demonstrated variations in the resulting Mooney images, it is important to note that they may not yet be optimal in terms of specific features. For example the global max edge technique might select a threshold to have as many edges as possible in the image. However, even more edges might be possible if they are detected on a local scale (*Weeks, Myler & Apley, 1994*). This means local thresholding techniques are far more flexible than global thresholding techniques and can produce a greater variety of thresholded and therefore Mooney images (*Weeks, Myler & Apley, 1994*). Even though, to our knowledge, local thresholding techniques were not used for Mooney image generation yet and generally perform worse in segmenting images (*Sezgin & Sankur, 2004*), we suggest that the application of local instead of global thresholding techniques might have the ability to further enhance or suppress certain features in Mooney images.

Additional research could furthermore remove some limitations of this study. For one, it could investigate the influence of image statistics on the interpretability of Mooney images. Are there any specific image configurations that make for ambiguous or unambiguous Mooney images? Additionally, here images were presented to each participant multiple times-each time under a different condition. This strategy was used to obtain repetitions within a participant-image pair. Future research could conduct similar experiments but without repeated image presentations to avoid the learning of concepts, and minimizing the influence of memory on task performance. However, to get sufficient data for each condition and image a larger pool of participants would be necessary.

In conclusion, this study demonstrated that Mooney images can be automatically created by pre-selecting a reasonable amount of smoothing and application of automatic thresholding techniques, achieving a high interpretability of the resulting Mooney images.

While the amount of smoothing influences the interpretability of the generated Mooney images slightly, the choice of thresholding technique has only minimal influence on interpretability, as long as a threshold somewhere around the images intensity mean is chosen. Future research could investigate whether local thresholding algorithms and the choice of template images might provide more influence on the interpretability of Mooney images.

## FOIL ANSWERS FOR CONCEPT DETECTION TASK

### Shape-related dimensions

The following shape related dimensions from the embedding of the THINGS dataset created by *Hebart et al. (2020)* were used to select shape-related concepts:

disc-shaped, course pattern, paper-related/text-related, long-thin, powdery/fine-scale pattern, spherical, repetitiveness, flat/patterned, thin/flat, stringy, has beams/support, has grating, cylindrical/conical.

### Increased task difficulty due to shape-related concepts

Selecting shape-related concept produced foil answers which were in terms of shape harder to discriminate from the correct answer. This can be seen in the examples presented in Table A1.

## GENERALIZED LINEAR MIXED MODELS

For further and more detailed data analysis of the concept detection task in experiment 2 we fitted a generalized linear (logistic) mixed model. We estimated the fixed effects of threshold selection method and size of the smoothing kernel as well as their interaction along with random effects for participant and template image on the probability to identify the concept in an image correctly. The model which assumes a Bernoulli process that depends on the mentioned factors using a logit link function can be described by the following lme4 formula:

```
correct ~ 1 + thresholding_technique +
    kernel_size + kernel_size:thresholding_technique
    (1 + thresholding_technique + kernel_size +
        kernel_size:thresholding_technique|participant) +
    (1 + thresholding_technique + kernel_size +
        kernel_size:thresholding_technique|original_img_name)
```

To evaluate whether kernel size and/or thresholding technique significantly influenced the probability of correctly identifying the concept we created three addition models. The formula of the second model which was the same as the first one but without an interaction between thresholding technique and kernel size was:

```
correct ~ 1 + thresholding_technique + kernel_size +
    (1 + thresholding_technique + kernel_size|participant) +
    (1 + thresholding_technique + kernel_size|original_img_name)
```

The third model did not take the kernel size as fixed effect into account. The corresponding formula is:

**Table A1 Comparison of foil answers selected randomly and by using shape similarity.**

| Correct answer | Random foils | Shape-related foils |
|---|---|---|
| Koala | Skateboard, seatbelt, latte | Weasel, warthog, chipmunk |
| Seagull | Trumpet, envelope, shirt | Badger, seal, bison |
| Olive | Doorbell, crutch, cardinal | Gumdrop, nut, almond |

```
correct ~ 1 + thresholding_technique +
    (1 + thresholding_technique|participant) +
    (1 + thresholding_technique|original_img_name)
```

The fourth model did not take the thresholding techniques as fixed effect into account. The corresponding formula is:

```
correct ~ 1 + kernel_size +
    (1 + kernel_size|participant) +
    (1 + kernel_size|original_img_name)
```

All effects were encoded using sum coding and were initialized with weakly informative priors, which were supposed to reduce the probability of unrealistic parameter values while not adding much bias to the model. Fixed effects coefficients and intercepts were given Normal (0, 1) priors. The priors of random effect standard deviations were Normal (0.5, 1) distributions which were truncated for values smaller zero as standard deviations cannot be negative. For the correlation matrices we used a LKJ (2) prior which was supposed to introduce a small bias against overly large correlations.

The posterior distributions of the models were estimates using a Markov Chain Monte Carlo procedure. We run four chains, each with 1,000 warmup-samples and 2,000 actual samples resulting in a total number of 8,000 post warmup draws from which the posterior was estimated.

## ACKNOWLEDGEMENTS

The authors thank Ruth Hartmann for feedback on this work and Martin Hebart for help with the THINGS dataset.

### Funding

Co-funded by the European Union (ERC, SEGMENT, 101086774). In addition, this work was supported by the Hessian Ministry of Higher Education, Research, Science and the Arts and its LOEWE research priority program 'WhiteBox' under grant LOEWE/2/13/519/03/06.001(0010)/77. The funders had no role in study design, data collection and analysis, decision to publish, or preparation of the manuscript.

### Grant Disclosures

The following grant information was disclosed by the authors:
European Union: ERC, SEGMENT, 101086774.

Hessian Ministry of Higher Education.
Research, Science and the Arts and its LOEWE research priority program 'WhiteBox': LOEWE/2/13/519/03/06.001(0010)/77.

## Competing Interests

The authors declare that they have no competing interests.

## Author Contributions

- Lars C. Reining conceived and designed the experiments, performed the experiments, analyzed the data, prepared figures and/or tables, authored or reviewed drafts of the article, and approved the final draft.
- Thomas S. A. Wallis conceived and designed the experiments, analyzed the data, authored or reviewed drafts of the article, and approved the final draft.

## Human Ethics

The following information was supplied relating to ethical approvals (*i.e.*, approving body and any reference numbers):

The experiments reported here were approved by the Technical University of Darmstadt Ethics Commission (Application number EK 77/2022).

## Data Availability

The data and code are available at Zenodo: Reining, L. C., & Wallis, T. S. A. (2024). Code and data: A psychophysical evaluation of techniques for Mooney image generation (1.0.0). Zenodo. https://doi.org/10.5281/zenodo.10714959.

The THINGS dataset and associated data are available at https://things-initiative.org.

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
