# Peer review of "A psychophysical evaluation of techniques for Mooney image generation"

_PeerJ, doi:10.7717/peerj.18059_

## Round 0.1 · original submission · Major Revisions

Dear Authors:

After peer review, we consider the paper needs revision. Please read all the comments below.

Best regards and thank you.

Dr. Manuel Jiménez

Reviewer 1 ·

Basic reporting

This paper explores various techniques for the automatic production of Mooney figure (black and white, thresholded versions of grayscale images). These images are widely used in visual neuroscience and computer vision as tests of visual processing. The images are typical generated by manually varying a threshold applied to selected images. Most images do not produce effective Mooney versions that are highly characteristic once recognized and are not overly difficult to recognize. This process is tedious and suffers from selection biases of the individuals who are creating the images. An automated approach would therefore be quite welcomed. Reining and Wallis take us through a carefully constructed set of algorithms for producing the images and a meticulous design for evaluating the recognition of the images.

The results show that there is no important difference between the thresholding algorithms and this is reassuring in a way – any methods will be as good as another so there is no motivation for using more time consuming techniques. They also show that less smoothing is better.

Experimental design

The experiments are clearly described. The production of the stimuli and the test procedure and analyses are all well defined and appropriate.

Validity of the findings

While the results are helpful to others in constructing Mooney figures for research purposes, I do have a concern about the recognition method the authors used. Participants were shown the test Mooney image and asked which of four possible things it might be (e.g., weasel, warthog, chipmunk, or koala for Figure 2). This cued recognition is vastly different from asking participants what they see and giving no cue. This classic naming task identified which images have sufficient cues, properly configured, to trigger recognition. When the authors provided a small set of possible matches, participants only need to look for specific distinctive features to get the answer correct (e.g., head shape or tail shape). An image that is correctly identified out of a set of 4 alternatives is not necessarily a useful Mooney image that could be used in other research.

The study would be much more compelling as a technical piece for the research community if the authors would include a pure naming task. Only images that achieve, say, 75% correct naming would be typically selected as acceptable Mooney images. The images generated by the different threshold algorithms and smoothing most likely fall more in the class of unrecognizable Mooney images that can be recognized if the original grayscale image is shown first. Providing the 4 response choices, one of which is correct, is a small step from providing the original image.

Additional comments

The authors need to categorize the type of images they are producing. Are they the Teufel (2018) kind that is initially meaningless but becomes a coherent percept once the original is seen, or are they of the easily and immediately interpretable kind? The 4 choice technique the authors used does not provide an answer here. The authors need to run a naming test to resolve this issue.

In addition, it would be good to have a link to all the images to get a better feel for what they look like. We only see the koala and the seagull in the paper and I could find any link to the generated images – I may have missed it.

·

Basic reporting

- I love this paper. Despite being a methods paper, it is clearly written, well-motivated, and positions itself well in the vision science and computer science literatures. Being able to straddle this intersection is not easy, but the paper interwove these literatures skillfully, such that it was a smooth read.

- I have some small methodological questions/clarifications, but my biggest concern is that I'm not actually sure what to make of the results in the end. It seems like smoothing is the technique that matters in the end — but why would this? Or is even this dependent ultimately on the original image? A great deal of background was laid out for the thresholding techniques, but not much for the smoothing techniques. So it would be helpful to understand *theoretically* why smoothing would change human visual perception performance.

- The criteria of what makes a good Mooney image (lines 74-76) was really helpful. I would encourage the authors to bring this back throughout the paper. For instance, when discussing spatial vs. non-spatial thresholding, it was unclear to me how either category would impact how the images would work under the criteria. It would be helpful to discuss predictions/hypotheses in the paragraph in line 158. (As noted above, same for the smoothing techniques.)

- It would strengthen the paper if the abstract included a little motivation as to how thresholding / smoothing techniques were sampled. Currently, it just says that they try out different techniques. The results start to speak to which techniques were tried, but by then they come out of nowhere. The question burning in the back of my mind while reading the abstract was whether and how diff. techniques would impact human visual perception *in theory*.

Experimental design

- In the memory recognition task, how were the two image foils selected there?

- Since the within-subjects design in Exp. 1 meant that participants effectively saw each concept four times, and it seems like this did this end up impacting people's difficult ratings / recognition performance eventually. Were the same foil concepts also used each time? So for each time they saw koala, it was always weasel, warthod, and chipmunk? This might also have contributed to the learning effect / participant strategies. Anyway, I appreciate the authors' being forthright about the limitations of their study in the end, but it might help to foreshadow these even earlier on? I'm left wondering about this as I'm reading the methods.

Validity of the findings

N/A

Additional comments

All-in-all, I look forward to seeing this in print. I think this would be a contribution to the literature, even if not entirely as satisfying as we would want. At least this will hopefully start the conversation for creating better Mooney images through which we can better study human visual perception.

---

## Round 0.2 · accepted · Accept

Dear Authors,

We are happy to let you know that your manuscript has been accepted for publication, pending the resolution of a few minor revisions as noted by our reviewer. We appreciate the thoroughness with which you have addressed the feedback provided during the review process and the quality of your work.

Minor Revisions Required in edition process:

1. Figure Adjustments:
• Figure 2: There is an extra parenthesis in the final box (Task 2) after the phrase “very difficult.” Please remove this to ensure clarity.
• Figure 6b: The word “rating” on the y-axis is not capitalised, while other titles use title case. To maintain consistency throughout your manuscript, please ensure that all figures use a consistent style—either title case or sentence case.
2. Mooney Image Sets:
• Please provide information on where the converted Mooney image sets can be accessed. This will be helpful for readers and researchers who may wish to replicate or extend your work.

Once these minor revisions have been made, please submit the final version of your manuscript.

Thank you for your contributions to the field.

Best regards,

Dr. Manuel Jiménez

Reviewer 1 ·

Basic reporting

Fine.

Experimental design

Fine.

Validity of the findings

Fine although of limited generalizability. The 4 choice recognition task does not identify the Mooney figures that would be used in research.

·

Basic reporting

Just a couple more minor comments re the figures + image set:

- Fig. 2: There is an extra parenthesis in the final box (Task 2), after "very difficult"

- Fig. 6b: The word "rating" in the y-axis is not capitalized, whereas title case is used in other previous instances (or maybe standardize everything to be sentence case - since sometimes authors label "Proportion correct" with sentence case, rather than title case). In any case, just make it consistent throughout

- Where can the converted Mooney image sets be accessed?

Experimental design

no comment

Validity of the findings

no comment